# The role of nuclear organization in *trans*-splicing based expression of heat shock protein 90 in *Giardia lamblia*

**Vinithra Iyer**[1], **Sheetal Tushir**[1], **Shreekant Verma**[2], **Sudeshna Majumdar**[3], **Srimonta Gayen**[3], **Rakesh Mishra**[2], **Utpal Tatu**[1]*

**1** Department of Biochemistry, Indian Institute of Science, Bangalore, India, **2** Centre for Cellular and Molecular Biology, Telangana, India, **3** Department of Molecular Reproduction, Development and Genetics, Indian Institute of Science, Bangalore, India

☯ These authors contributed equally to this work.

* tatu@iisc.ac.in

**Data Availability Statement:** All relevant data are within the manuscript and its Supporting information files.

## Abstract

Hsp90 gene of *G. lamblia* has a split nature comprising two ORFs separated by 777 kb on chromosome 5. The ORFs of the split gene on chromosome 5 undergo transcription to generate independent pre-mRNAs that join by a unique *trans*-splicing reaction that remains partially understood. The canonical cis-acting nucleotide elements such as 5'SS-GU, 3'SS-AG, polypyrimidine tract and branch point adenine are present in the independent pre-mRNAs and therefore *trans*-splicing of Hsp90 must be assisted by spliceosomes *in vivo*. Using an approach of RNA-protein pull down, we show that an RNA helicase selectively interacts with HspN pre-mRNA. Our experiments involving high resolution chromosome conformation capture technology as well as DNA FISH show that the *trans*-spliced genes of *Giardia* are in three-dimensional spatial proximity in the nucleus. Altogether our study provides a glimpse into the *in vivo* mechanisms involving protein factors as well as chromatin structure to facilitate the unique inter-molecular post-transcriptional stitching of split genes in *G. lamblia*.

## Author summary

*Giardia lamblia* causes the most common enteric disease called Giardiasis worldwide in humans and animals. *Giardia* is an intriguing model organism to study molecular evolutionary processes owing to its peculiar position, at the transition of prokaryotes and eukaryotes. Previous studies from our lab have shown a unique mode of expression of the Hsp90 gene, which is fragmented into two halves and remotely spaced in the *Giardia* genome. We showed that pre-mRNAs arising from the distant genes undergo molecular stitching by *trans*-splicing to generate the mature message. This process of molecular stitching of Hsp90 at the RNA level is unique to *Giardia* and is ill explored. The current study sheds light on the mechanisms of this molecular jugglery unique to this neglected parasite. In this study, we describe the role of nuclear architecture in bringing the fragmented genes, which are otherwise far apart, in close proximity to facilitate their molecular stitching. In addition, we highlight the role of protein factors in orchestrating this

**Funding:** This research was supported by Department of Biotechnology (Grant No.- DBTO-0403) and DBT-IISc Partnership grant to S.G., R.M and U.T.; V.I. and S.T. acknowledge GATE fellowship from IISc; S.V. acknowledges financial support from CCMB; S.M. was supported by scholarship from IISc. The funders had no role in study design, data collection and analysis, decision to publish, or preparation of the manuscript.

**Competing interests:** The authors have declared that no competing interests exist.

**Abbreviations:** 3C, Chromosome Conformation Capture; Dhcβ, Dynein heavy chain beta; Dhcγ, Dynein heavy chain gamma; FA, Formamide; FISH, Fluorescence *in situ* hybridization; FLRNA, Full length RNA; gDNA, Genomic DNA; Hsp90, heat shock protein 90; MS, Mass Spectrometry; nt(s), nucleotide(s); NTPs, nucleotide triphosphates; RBP (s), RNA Binding protein(s); SSC, Saline Sodium Citrate; Ts, *Trans*-splicing.

molecular feat. Our results point to the unique mechanism(s) which can serve as potential targets to develop specific treatments against this important pathogen.

## Introduction

*Giardia lamblia* is a human and animal protozoan parasite that causes the most common enteric disease called Giardiasis worldwide. Giardiasis, commonly known as the traveller's diarrhoea is widespread in the developing and underdeveloped countries where poor sanitary conditions prevail [1].

As an organism, *Giardia lamblia* is a deep branching eukaryote and thereby interesting model to examine evolutionary processes. *Giardia* genome was sequenced in 2007, which revealed that the parasite genome is intron poor with only 6 cis-spliced genes identified [2–5].

Our lab has previously shown that the Hsp90 gene of *Giardia lamblia* gets expressed from two ORFs dispersed by 777 kb on chromosome 5. The two ORFs generate individual pre-mRNAs and undergo canonical GU-AG based post transcriptional repair to generate mature full length Hsp90 RNA by *trans*-splicing [6]. Subsequently, two other genes, namely Dhcβ and Dhcɣ were also reported to get expressed by a similar *trans*-splicing based mechanism in *G. lamblia* [7–9]. This unique process of post-transcriptional stitching of independent pre-mRNAs is distinct from the spliced-leader (SL) *trans*-splicing (SLTS) observed in trypano-somes, nematodes, euglenozoans, perkinsozoans and dinoflagellates. It involves linking of 5' capped short leader sequences (16–52 nts) originating from a different location in the genome to individual RNAs in the poly-cistronic mRNAs, in the process making them readable by the translational machinery to give rise to the functional protein [10,11].

*Giardia* contains functional spliceosome. The snRNAs, U1, U2, U4, U5 and U6 have been identified bioinformatically and their expression has been confirmed experimentally [12,13]. The identified snRNA candidates in *Giardia* display similar core structural features as compared to the snRNAs of higher eukaryotes but have structural simplification [12]. *In silico* analysis showed that *G. lamblia* retained the more conserved major snRNA-associated spliceosomal proteins and the less conserved ones might not have existed or could have been lost [14].

Sequence analysis of HspN and HspC pre-mRNA substrates showed the presence of cis-sequence elements; GU-AG intron-exon boundary elements, polypyrimidine tract as well as branch point adenine in Hsp90 pre-mRNAs[15,16]. The presence of the canonical sequence elements in the independent pre-mRNAs and the presence of genes coding for functional min-imal spliceosomal apparatus in the *Giardia* genome suggested that the post-transcriptional stitching of HspN and HspC pre-mRNAs may be assisted by spliceosomes.

With a view to examine the splicing process *in vivo* and examine specific nuclear factors, we have analysed the interacting proteins by RNA-protein pull down assay. Using this approach followed by MS based protein identification, we found a putative RNA helicase from *Giardia* nuclear extract associating with HspN pre-mRNA. In this study, we also address the *in vivo* mechanisms, by studying the nuclear organization of *trans*-spliced genes and identifying nuclear factor(s) that may aid this unique process. In terms of nuclear organization, our chro-mosome conformation capture (3C) as well as DNA FISH results, showed split genes of Hsp90 to be in close proximity in the *Giardia* nucleus. Altogether, our study provides a glimpse into the *in vivo* mechanisms involving nuclear organization as well as nuclear factors operating to support this novel, *trans*-splicing-based expression of Hsp90 in a deep branching eukaryote.

## Methods

### Nuclear extract preparation

*Giardia* Nuclear extract was prepared with minor modifications in the protocol described by Janet Yee *et al.*[17].

### *In vitro* transcription of pre-mRNAs devoid of MCS

Construct harbouring HspN, HspC, HspNΔ26, HspCΔ27 and FLHsp90 were linearized using EcoR1 and were used as a template for PCR reaction using primers (forward primers) having T7 promoter sequence as described in S2 Fig. These amplified products were extracted from agarose gel using MN gel and PCR clean-up kit. The amplified gel-purified DNA devoid of MCS sequence in the vector was used as a template for the synthesis of respective biotinylated RNAs using T7 RNA polymerase with biotin-16-UTP (Roche) and UTP in the ratio 1:1. All these *in vitro* synthesized RNAs are truncated, retaining intact split intronic sequences but lacking MCS sequence present in the construct. RNA integrity was checked on 1.5% MOPs Formaldehyde Agarose gel (S3 Fig).

### RNA-protein pull down

RNA-protein pull down assay was performed according to P. Dave *et al.* [18]. To identify the interacting nuclear protein partners of pre-mRNAs, 500 μg of *Giardia* nuclear extract was incubated with 15 μg of biotinylated RNAs for 3 hrs at 4°C with continuous mixing on end-to-end rotor. Proteins binding and interacting with biotinylated RNA were pulled down by incubating for 1.5 hrs with 25 μl of streptavidin magnetic beads (Roche). Non-specifically bound proteins were eluted by 3 vigorous washes for 10 minutes, each with 300 mM–600 mM NaCl. Specific RNA binding proteins interacting were dissociated from RNA bound to streptavidin magnetic beads by boiling in SDS loading buffer, resolved on 8% SDS-PAGE, and visualized by silver staining.

### MS sample preparation and analysis

1. ***In-gel digestion***-Specific bands above 100kDa interacting with *HspN*, *HspC*, *HspNΔ26* and *HspCΔ27* (lanes 1 to 4) were sliced out from stained SDS-PAGE gel and processed further using standard in-gel digestion protocol. Specific bands were further sliced into fine gel plugs and destained with 30mM potassium ferricyanide and 100 mM sodium thiosulphate in a 1:1 ratio. The gel pieces were then subjected to wash with a solution having 50 mM ammonium bicarbonate (Sigma-Aldrich) in 50% acetonitrile (ACN). This step was repeated till the gel plugs became transparent. Further gel plugs were reduced using 10 mM DTT (Sigma-Aldrich) in 50 mM ammonium bicarbonate at 56°C for 45 min followed by alkylation with 55 mM iodoacetamide in 50 mM ammonium bicarbonate at 37°C for 30 min in the dark. The gel pieces were then vacuum dried and in-gel digestion was carried out with 20 ng/μl trypsin (Promega) in 50 mM ammonium bicarbonate overnight at 37°C. Peptides were extracted using a solution containing 5% formic acid (MS grade) in 60% ACN. The supernatant of all the samples were collected, subjected to vacuum dry and stored at -20°C.

2. ***Mass spectrometry and database search***-The dried trypsin digested peptides were reconstituted in a mixture of 70% ACN and 30% MilliQ water containing 0.1% formic acid. The protein digests were analysed using Agilent 1290 Infinity II LC system coupled with Agilent Advance Bio Q-TOF (6545XT). The column used for chromatography was Agilent AdvanceBio Peptide Map (2.1x 150 mm, 2.7μ). Mobile phase A was 0.1% formic acid in MilliQ

water and mobile phase B was ACN (in 0.1% formic acid). The peptides were separated by using a 35 min gradient flow at a flow rate of 0.4 ml/min. The MS and MS/MS scan were acquired in the positive ion mode and stored in centroid mode. The V cap was set at 3500V. The drying gas flow rate and the temperature were set at 13 litres/min and 250˚C, respectively. Collision energy with a slope of 3.6V/100 Da and an offset of 4.8V was used for fragmentation. The precursor ion data were captured in a mass range of 300–1700 m/z and product ions data were acquired in the range 50–1700 m/z. Reference exclusion was given for 0.05 min after 1 spectrum. The raw data obtained was analysed using MaxQuant software. For identification of peptides, they were searched against *Giardia lamblia* proteome in Uniprot (Proteome ID- UP000018040) with cysteine carbamidomethylation as a fixed modification and methionine oxidation and protein N-term acetylation as variable modifications.

## Preparation of *Giardia* nuclei for chromosome conformation capture (3C)

$1 \times 10^8$ axenically growing log phase *Giardia* trophozoites were divided equally into two microtubes such that each tube now has $0.5 \times 10^8$ cells. Trophozoites in each tube were then washed with 1x phosphate buffered saline (PBS). One of the tubes was then used for (1.5%) formaldehyde fixation in the extraction buffer. Fixation was conducted for 20 min at 300 rpm. Both the tubes containing equal number of cells were then treated with equal volume of 1.25 M glycine supplemented extraction buffer containing mild detergent (NP-40) and incubated for 8–10 mins on ice with gentle mixing at intervals. The entire content was then introduced upon a cushion of cold 0.8 M sucrose containing extraction buffer components [19]. Nuclei were recovered in the pellet post centrifugation at 6000 rpm for 20 min at 4˚C. Nuclei enriched in the pellet was washed thrice with extraction buffer without NP-40. The control and the formaldehyde cross-linked nuclei were then aliquoted equally into five microtubes and stored at -80˚C until further use.

## Preparation of 3C library

Purified formaldehyde cross-linked nuclei were subjected to washes with 500 μl of 1.2x Restriction Enzyme (RE) buffer twice. This step was followed by treatment with 0.1% SDS at 65˚C for 30 min and then treatment with 1% triton-X-100. 100 μl of sample from the tube was kept aside for quality check of intact chromatin at the end of the procedure. 100 μl of 1.2x RE buffer was introduced into the tube and chromatin subjected to restriction digestion with 1000 U of EcoRI restriction enzyme at 37˚C overnight. The reaction volume was diluted 4 folds with nuclease free water. Restriction enzyme was then inactivated with 0.1% SDS at 65˚C for 30 min followed by treatment with 1% triton-X-100 to inactivate SDS. 200 μl of sample was aspirated in fresh tube and stored at -20˚C to examine the quality of the digested chromatin at the end of the procedure. Ligation of the diluted chromatin was performed using T4 DNA ligase in T4 DNA ligase buffer containing 1x BSA at 16˚C for 6–8 hrs. Ligation step of the diluted chromatin would promote intramolecular ligation instead of intermolecular ligation and would result in 3C library. Post ligation decrosslinking of the 3C library was performed by treatment with 20 μg/ml of proteinase k for 10 hr at 65˚C at 700 rpm. This was followed by RNase A treatment for 1 hr at 37˚C and extraction of chromatin by phenol:chloroform method. The chromatin was precipitated using 1/10ᵗʰ volume of ammonium acetate and 2.5 volume of chilled molecular biology grade ethanol in the presence of glycogen (20 μg/ml). Chromatin was then washed with 70% ethanol, dried and dissolved in nuclease free water. The

3C library procedure was performed as described by Dekker *et al.*, with certain modifications described above [20]. The concentration of intact, digested and ligated cross-linked chromatin was estimated by nanodrop. Quality check of intact, digested and ligated crosslinked chromatin was performed on 0.8% agarose gel as shown in S5B Fig.

### Preparation of un-crosslinked control gDNA

gDNA was purified from un-crosslinked control chromatin. Purified gDNA was subjected to digestion as described above. However, digested gDNA was not subjected to dilution [20]. Digested purified gDNA was directly proceeded for ligation to allow random ligation events unlike the dilution step before ligation in the protocol for 3C library preparation which promotes intramolecular ligation. Subsequent steps were identical to the procedure for 3C library preparation. The concentration of intact, digested and ligated control un-crosslinked gDNA aspirated at the end of different steps was determined by nanodrop and the quality examined on 0.8% agarose gel as shown in S5B Fig.

### Designing primers for 3C

EcoR1 loci were chosen on chromosome 5 and chromosome 3; some in the vicinity of HspN and HspC ORFs on chromosome 5, some in between HspN and HspC, few close to Dhcβ C-2 on chromosome 5, several in between HspN and Dhcβ C-2 and seven loci in the vicinity as well as one within Dhcγ C-1 ORF on chromosome 3. Forward and reverse primers were designed approximately 100–150 bps away from and flanking each of these loci. Primer pairs for each loci were designed using thermofisher Tm calculator setting parameters for taq DNA polymerase. The melting temperatures (Tms) of each of the primer pairs were approximately 60˚C. The primer pairs designed were further analysed for their inability to form self-dimer, primer dimer and hair pin loops using IDT's oligo analyser software. The primer sequences for each of the different loci used in the 3C are listed in the supplemental excel file (S1 Data).

### Preparation of template to check primer efficiency

The forward and reverse primers for each of the different loci were used to enrich the corresponding loci by PCR from *Giardia* genomic DNA. The PCR products were resolved on 1.2% agarose gel and the bands obtained from different loci were gel eluted using MN PCR and Gel Clean up kit. Concentration of different eluted PCR products were estimated by nanodrop and equal concentration of eluted products were mixed together. Mixture of equal concentrations of different enriched loci was then digested with EcoR1 overnight. Post digestion, the restriction enzyme was inactivated with 0.1% SDS at 65˚C for 20–30 mins at 700 rpm. This was followed by inactivation of SDS with 1% Triton-X-100 at 37˚C for 1 hr. The pool of different digested enriched loci was then ligated using T4 DNA ligase in ligase buffer without dilution for 6–8 hours at 16˚C. During this ligation step each digested fragment of a particular loci would have equal opportunity to get ligated to the fragments of all the other loci. The mixture of randomly ligated loci was then extracted with equal volume of phenol:chloroform (1:1). DNA was precipitated using 1/10th volume of ammonium acetate and 2.5 volumes of chilled absolute ethanol followed by 70% ethanol wash. The DNA pellet was dried, and the concentration was estimated by nanodrop and this mixture of randomly ligated loci was used as the template to check primer efficiency. 2F primer was used as the constant primer while varying other forward primers.

### DNA fluorescence *in situ* hybridization (DNA FISH)-

DNA FISH with double-stranded probes was performed according to [21,22]. Probes of the accurate sizes were first PCR amplified from *Giardia* genomic DNA using Phusion polymerase, gel extracted and then used as a template for the labelling reaction. The dsDNA FISH probe was synthesized by random priming using BioPrime DNA Labelling System (Invitrogen, #18094–011). The HspN and Enolase probe was labelled with Cy3-dUTP (ENJO) and other probes including- HspC, Protein 21.1, Dhcβ C-2, Dhcβ C-3, Dhcγ C-1 and CWP1 probes were labelled with FITC-dUTP (Roche) using a Phusion amplified PCR product as a template. Labelled probes from multiple templates were precipitated in a 3M sodium acetate solution along with 200 μg of yeast tRNA (Invitrogen, #15401–029) and 150 μg of sheared, boiled salmon sperm DNA (Invitrogen, #15632–011). The solution was then spun at 15,000 rpm for 30 min at 4˚C. The pellet was washed consecutively with 70% ethanol and 100% ethanol while spinning at 15,000 rpm for 4 min at room temperature. The pellet was dried and resuspended in deionized formamide (VWR, # 0606–500). The probe was denatured by incubating at 90˚C for 10 min followed by an immediate 5 min incubation on ice. A 2X hybridization solution consisting of 4X SSC and 20% Dextran sulfate (Millipore, #S4030) was added to the denatured solution. All probes were stored in the dark at −20˚C until use.

For DNA FISH, *Giardia* cells were cultured on gelatin-coated glass coverslips. The cells were then permeabilized through sequential treatment with ice-cold cytoskeletal extraction buffer (CSK:100 mM NaCl, 300 mM sucrose, 3 mM MgCl$_2$, and 10 mM PIPES buffer, pH 6.8) for 30 sec, ice-cold CSK buffer containing 0.4% Triton X-100 (Fisher Scientific, #EP151) for 30 sec, followed by ice-cold CSK for 30 sec. After permeabilization, cells were fixed by incubation in 4% paraformaldehyde for 10 min. Cells were then rinsed thrice in 70% ethanol and stored in 70% ethanol at −20˚C prior to DNA FISH.

Following permeabilization, the cells on coverslips were dehydrated through 2 min incubations in 70%, 85%, 95%, and 100% ethanol solutions and subsequently air-dried. RNase treatment was carried out for 45 min at 37˚C followed by another cycle of dehydration. The cells were denatured in 70% formamide/2X SSC on a glass slide at 95˚C for 11 min, which was immediately followed by -20˚C ethanol dehydration series and then air-dried. The coverslips were then hybridized to the probe overnight in a humid chamber at 37˚C. The samples were then washed thrice for 7 min each at 37˚C with 2X SSC/50% formamide, 2X SSC, and 1X SSC. A 1:250,000 dilution of DAPI (Invitrogen, #D21490) was added to the third 2X SSC wash. Coverslips were then mounted on slides in Vectashield (Vector Labs, #H-1000).

Images were captured and visualised by Laser Confocal microscope (Olympus, FLUO-VIEW FV3000, Japan). Images were analysed for co-localization of signals by using the JACoP plugin of the ImageJ software, as described previously (Bolte and Cordelieres, 2006). Mander's co-localization coefficient, which denotes the fraction of overlap of one color over the other, was chosen as one of the parameters of analysis. The value obtained for each cell was plotted in the form of a box and whisker plot, denoting mean with standard deviation.

## Results

### A DEAD/H box RNA helicase interacts with HspN

Our previous study showed that HspN and HspC pre-mRNAs harbouring the canonical sequence elements, 5'SS-GU, 3'SS-AG, branch point adenine and polypyrimidine tract could undergo *trans*-splicing *in vitro* without any nuclear protein factors, with the help of complementary sequence elements in the pre-mRNA substrates. However, the self-splicing reaction

was highly inefficient and could only be detected using indirect approaches like RT-PCR, Nanostring technology and northern blot [23].

With a view that Hsp90 *trans*-splicing reaction, *in vivo* must be spliceosomal dependent, we embarked on identifying potential RNA binding proteins (RBPs) that may facilitate the *trans*-splicing reaction *in vivo*.

We used an approach of RNA-protein pull down to identify potential nuclear factors that may interact with pre-mRNAs in *G. lamblia* nuclear extracts. As outlined in Fig 1A, we *in vitro* synthesized biotin-16-UTP body labelled partial pre-mRNAs; *HspN*, *HspNΔ26*, *HspC*, *HspCΔ27*, Hsp90 FLRNA as described in supporting information (S2 Fig). Nuclear extract was prepared from axenically grown *Giardia* trophozoites as described in Methods [17]. In order to identify the RBPs which may bind and interact with the pre-mRNAs, we incubated 15 μg of biotin-16-UTP body labelled pre-mRNAs with 500 μg of *Giardia* nuclear extract for 3 hours at 4˚C. 25 μg of streptavidin coated magnetic beads were employed to pull down the RBPs bound to biotinylated pre-mRNAs. A schematic of RNA-Protein pull down assay is illustrated in Fig 1A. Proteins pulled down with different pre-mRNAs were resolved on 8% SDS PAGE and visualized by silver staining (n = 3) (Lanes 1–4, Fig 1B). Lanes 5 and 6 in Fig 1B display pull down profiles with FLRNA (devoid of intronic sequences) and Beads alone (BA), which serve as negative controls for our assay.

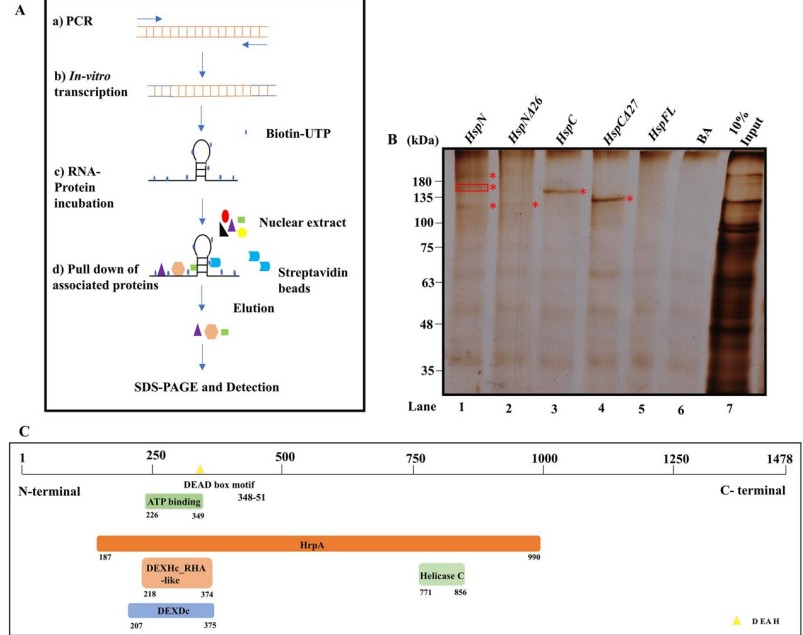

**Fig 1. DEAD box RNA helicase interacts with *HspN* and may facilitate base pairing interaction with *HspC* complementary sequence.** A) Schematic representation of RNA-protein pulldown experiment. B) Silver stained 8% SDS-PAGE showing specific nuclear proteins pulled—down with biotinylated pre-mRNAs- HspN, HspNΔ26, HspC, HspCΔ27 (Lanes 1–4). Biotinylated FL Hsp90 and bead alone (BA) in lanes 5 and 6 serve as negative controls. 10% input (lane 7) shows proteins in the nuclear extract. Unique bands obtained after the RNA-Protein pull down assay, marked in the Figure, were cut, and processed further for mass spectrometric identification of RNA binding proteins. ATP dependent RNA helicase DHR1 was identified from the middle intense band highlighted explicitly using a box, among the three intense bands pulled down with wild type *HspN* in Lane 1. Other unique proteins pulled down with *HspN* (lane 1) in the other two bands, proteins pulled down with *HspNΔ26*, *HspC* and *HspCΔ27* are subject matter for future research. C) Schematic representation describing the different domains of *Giardia* RNA helicase identified to be interacting with HspN pre-mRNA. The protein-protein blast with pre-mRNA splicing factor, ATP dependent RNA helicase- PRP16 of *S. pombe* revealed the presence of DEAD/H box helicase motif between 348–351 aa which was previously not annotated.

The unique bands of molecular size above 100 kDa pulled down in lanes 1–4 (Fig 1B) were cut and processed for protein identification by mass spectrometry analysis along with the corresponding region of negative control lanes (5 and 6). Among the proteins identified by LC-MS/MS, from the middle intense band highlighted in lane 1 (Fig 1B) we found an ATP dependent RNA helicase DHR1 (GL50803_17387) interacting with wild type HspN pre-mRNA (n = 3). This RNA helicase interacts selectively with HspN pre-mRNA as it was not identified from the corresponding region of other lanes including deleted construct and negative controls. We identified other unique RNA Binding proteins (RBPs) in addition to spliceosomal protein, U2 small nuclear ribonucleoprotein A from the bands highlighted in lanes 1–4 of Fig 1B.

Fig 1C shows different domains in the putative ATP-dependent RNA helicase DHR1, of *Giardia lamblia*. We performed a protein-protein blast to identify conserved domains in this protein. Amino acids (aas) between interval 187–990 showed specific hits to HrpA superfamily like helicases. We found specific hits for DEXDc and DEXHc RHA like family proteins with N- terminal ATP binding site and helicase conserved C-terminal domain. In addition, we were able to locate a conserved sequence of DEAD/H box helicase motif between 348–351 aa. Overall, our protein-protein blast result and domain alignment showed similarity of DEAD/DEXH-box containing *Giardia* ATP-dependent RNA helicase similar to pre-mRNA splicing factor PRP16 of *S. pombe*, which is known to be involved in the fidelity of branch-point recognition in yeast splicing [24]. The RNA helicase may augment the process of initial juxta-positioning of Hsp90 pre-mRNAs essential for accurate *trans*-splicing reaction.

Clearly, multiple accessory factors will be involved in facilitating this unique post-transcriptional repair. This study lays the foundation for future detailed studies on specific RBPs like RNA helicase, how they may interact with *HspN in vivo* and mechanisms to unwind the secondary structure of interacting pre-mRNA(s) to support the *trans*-splicing process. It is possible that the RNA helicase may be of general nature, also interacting with other RNAs in the nucleus and hence, further biochemical experiments are required to understand the mechanistic aspects of this protein.

## Nuclear proximity of HspN and HspC

HspN and HspC ORFs are separated by 777 kb sequence on chromosome 5. Despite the linear distance between the genes, the independent pre-mRNAs transcribed from these genes acquire proximity and *trans*-splice to give rise to mature mRNA. We hypothesized that chromatin looping may bring the two ORFs in proximity to aid this novel post-transcriptional repair.

To investigate whether HspN (GL50803_98054) and HspC (GL50803_13864) ORFs are in physical proximity in the nucleus, high resolution Chromosome Conformation Capture (3C) technology was employed. *Giardia* trophozoites were subjected to formaldehyde cross-linking to fix the protein-DNA and protein-protein interactions. Equal number of trophozoites not treated with formaldehyde served as un-crosslinked control. Nuclei was isolated from formaldehyde cross-linked and un-crosslinked cells. Treatment of fixed nuclei with 0.1% SDS ruptured the nuclear membrane and the chromatin was then subjected to digestion with EcoRI followed by dilution and ligation to promote intramolecular ligation which would enrich ligation of interacting loci in the fixed chromatin. Un-crosslinked gDNA was prepared as described in Methods and was employed as control. Simplified schematic for the 3C workflow followed is shown in S5 Fig. Quality of the cross-linked and control un-crosslinked gDNA after each step was checked at the end of the procedure on 0.8% agarose gel as shown in S5B Fig. Lanes 1 and 4 show intact un-crosslinked gDNA and fixed chromatin. Lanes 2 and 5

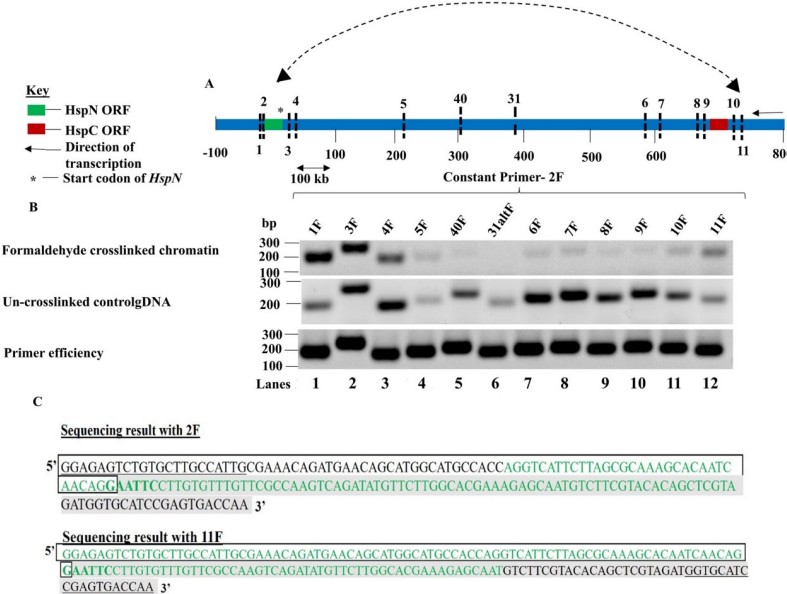

**Fig 2. HspN and HspC ORFs physically interact in the nucleus.** A) Schematic representation of the relative positions of different EcoR1 loci in the vicinity and in between HspN and HspC ORFs on chromosome 5. B) First panel shows the PCR products obtained with constant primer 2F with other forward primers designed across the different loci in vicinity and in between HspN and HspC ORFs chosen for this study. Loci 1 and 3 which are in proximity to locus 2 show intense bands of corresponding sizes. As the distance from locus 2 increases, the PCR amplicons fade out and the PCR signal then starts picking up from locus 6 gradually; increasing significantly at locus 11 confirming physical proximity and interaction between loci 2 and loci 11 which are proximal to HspN and HspC ORFs respectively. Second panel shows the PCR products obtained with constant 2F primer with other forward primers corresponding to the different chosen loci for randomly ligated control gDNA. Third panel shows PCR amplicons of equal intensities with 2F constant primer and other forward primers indicating equal primer efficiencies. C) The enriched hybrid formed from locus 2 and locus 11 (Fig 2B, panel 1-lane 12) was excised and subjected to Sanger's DNA sequencing with primers; 2F and 11F. Sequencing result with 2F as the primer could detect the presence of sequences from locus 2 and locus 11 which confirms the presence of this hybrid in the enriched band. Similarly, sequencing result with 11F as the primer could detect the presence of sequences from locus 11 and locus 2 which confirms the presence of the hybrid in the enriched band.

show smeared digested control gDNA and fixed chromatin. Lanes 3 and 6 display randomly ligated control gDNA and 3C library, respectively.

To validate the interaction between HspN and HspC ORFs, we chose 13 different EcoRI loci, 4 in the vicinity of each of these ORFs and 5 loci distributed in between the two ORFs as shown in Fig 2A and S8 Fig. Primers were designed flanking these EcoRI recognition sequences (S9 Fig). The EcoRI loci refers to the EcoRI recognition sites in the vicinity and between HspN and HspC ORFs investigated to determine physical proximity between the ORFs (S8 Fig). The distance of the different recognition sites from the start codon of HspN has also been elaborated in S8 Fig.

Semi-quantitative polymerase chain reaction (PCR) was employed for the 3C library obtained from fixed chromatin and randomly ligated gDNA from un-crosslinked chromatin with constant primer 2F and varying the primers from other loci in each reaction. Fig 2B, first panel shows PCR products obtained from 3C library of fixed chromatin and second panel displays the amplified products from control un-crosslinked randomly ligated gDNA as described in S6 Fig.

The high intensity of the bands obtained for loci close to locus 2 in first panel indicated the linear proximity of these loci. However, the intensity of the amplicons fades out as the distance

in kbs increases from locus 2 and begins to increase gradually at loci 6 and 7. As the distance increases further, a significant increase in amplicon intensity was observed when 2F was the constant primer and 11F was used as the variable primer. This was not because of differing primer efficiencies as 2F displays equal primer efficiency with primers for other loci (Fig 2B, panel 3).

Thus, our results suggest that dilution of cross-linked chromatin promotes ligation of interacting loci (in the vicinity of HspN and HspC). Therefore, the 3C library is enriched in interacting locus 2 and loci 11 hybrid. The relative cross-linking frequency was plotted from 3 independent experiments as a function of distance in kb from locus 2 as shown in S10 Fig.

On the other hand, the un-crosslinked control gDNA shows randomly increasing and decreasing intensity of amplicons as the distance from HspN increases due to random ligation (Fig 2B, middle panel). Thus, our 3C result provides experimental evidence for co-localization of HspN and HspC ORFs in the three-dimensional nuclear milieu which may facilitate one pre-mRNA to seek for the other pre-mRNA on the basis of 26 nucleotide complementary sequences to ensure specificity and accuracy of Hsp90 *trans*-splicing reaction *in vivo*.

## Fluorescence *in situ* hybridization confirms co-localization of HspN and HspC in *Giardia* nucleus

To further confirm the 3C result, we resorted to a direct microscopic approach to visualize proximity of HspN and HspC genes by DNA Fluorescence *in-situ* hybridization (DNA FISH). We used fluorescently labelled DNA probes designed against HspN and HspC ORFs using primers as listed in the Supplemental excel file (S1 Data). We performed DNA FISH with probes against HspN and HspC loci to investigate their nuclear proximity, whereas for control we used Cy3 and FITC labelled probes against Enolase gene (GL50803_11118) and Protein 21.1 (GL50803_24412), respectively. Both the control genes are located in between HspN and HspC on chromosome 5. DNA FISH was performed as described in the Methods section. Multiple "Z stacks" were captured for every image, and the resulting images were stacked to give the "merged" image. Fig 3A shows that both HspN and HspC genes (represented as red and green signals from Cy3 and FITC labelled probes, respectively) co-localize with each other. On the other hand, the fluorescent signals from Cy3 labelled probe for Enolase gene and FITC labelled probe for Protein 21.1 on the same chromosome and in between HspN and HspC ORFs, do not co-localize in the nucleus and therefore served as a negative control. In all the cases, images were analysed for co-localization of signals by using the JACoP plugin of the ImageJ software, as described previously [25].

Fig 3B depicts a Coste's mask depicting true co-localization of the signals as a white dot (merge of red and green foci). The white dot was observed with fluorescent probes against HspN and HspC. However, no co-localisation was observed in control (left panel), as shown in Fig 3B.

We analyzed at least 30 cells for quantitation of co-localization between signals in *Giardia* nuclei. Mander's co-localization coefficient, which denotes the fraction of overlap of one colour over the other, was chosen as the parameter for analysis. Mander's coefficient represents values in the range of 0–1, where 0 signifies no co-localization and 1 signifies 100% co-localization, which was evaluated using JACoP plug-in of ImageJ software. Fig 3C displays the Mander's coefficient highlighting the extent of co-localization of HspN and HspC. The box and whisker plot demonstrates the fraction of red signal overlapping the green signal (red) as well as the fraction of green signal overlapping the red signal (green) in both experiment as well as control. Our quantitation results suggest that the mean co-localization frequency for HspN and HspC is 0.75 among the 30 cells analyzed. Control refers to co-localization analysis

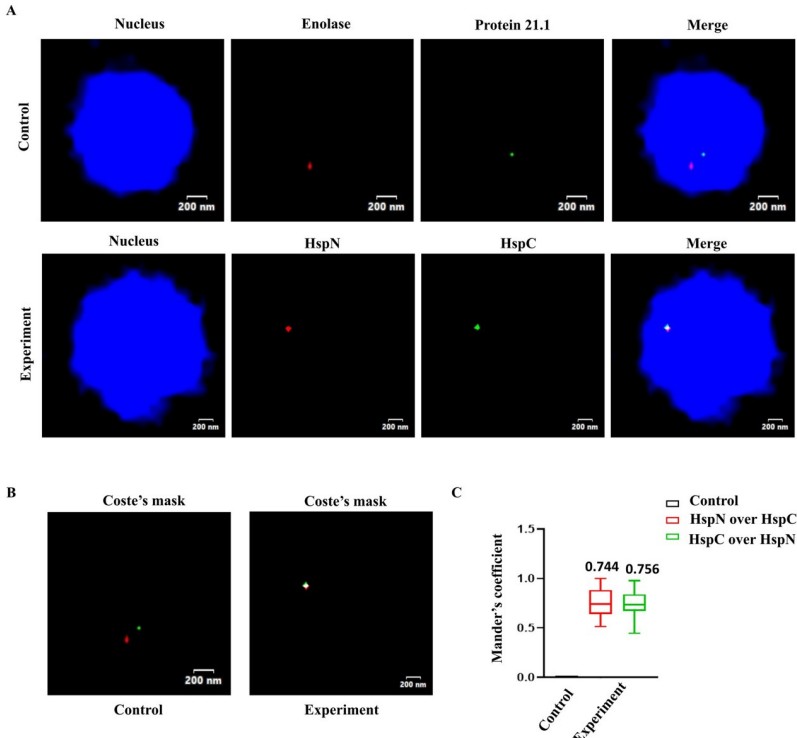

**Fig 3. Fluorescence *in situ* hybridization shows co-localization of HspN and HspC ORFs within the nucleus.** A) Representative two-coloured FISH images showing localization of HspN and HspC loci. Top panel shows FISH using probes against Enolase (red) and Protein 21.1(green) which serves as a negative control for co-localization. Bottom panel represents co-localization of HspN (red) and HspC (green) loci. Multiple "Z stacks" were captured for every image in control as well as experiment and the resulting images were stacked to give the "merged" image. Genomic DNA was counterstained with DAPI (blue). B) Depicts the Coste's mask (merge of green and red foci) for representative FISH images. Merged foci (white dot) as shown in experiment depicts true co-localization of the HspN and HspC loci. C) Quantitation of co-localization signal in *Giardia* nuclei. Box-and-whisker plots depicting Mander's co-localization coefficient (range 0–1, where 0 signifies no co-localization and 1 signifies100% co-localization) as evaluated by JACoP plug-in of ImageJ software. A minimum of 30 cells were analysed for co-localization of red and green signals and plotted as the fraction of red overlapping green (red) and green overlapping red (green). Control refers to co-localization analysis performed with Enolase and Protein 21.1 probe.

performed with Enolase and Protein 21.1 probe showing Mander's co-localization coefficient of zero. Therefore, our DNA FISH result confirms that HspN and HspC are in physical proximity despite several kbs of linear distance between them.

## Chromatin structure facilitates proximity of different *trans*-spliced genes in *Giardia* nucleus

We hypothesized that similar chromatin looping mechanism may be instrumental in bringing different *trans*-spliced genes in close proximity and there may be a specialized region in *Giardia* nucleus where all the *trans*-spliced genes may be concentrated.

Therefore, we performed 3C to determine whether HspN is in proximity to other split genes; Dhcβ C-2 (GL50803_10538) on the same chromosome and Dhcγ C-1 (GL50803_16804) on chromosome 3. We prepared 3C library from formaldehyde cross-linked and randomly ligated gDNA from un-crosslinked trophozoites as described previously.

We chose different EcoRI loci to investigate physical proximity between HspN and Dhcβ C-2 (S11 Fig). Primers were designed flanking these EcoRI recognition sites as shown in S12 Fig. The distance of the different recognition sites from the start codon of HspN has also been elaborated in S11 Fig.

Semi-quantitative polymerase chain reaction was performed using formaldehyde fixed 3C library and randomly ligated gDNA control from un-crosslinked trophozoites with constant primer 2F and varying the primers from other loci in each reaction. Fig 4B, first panel shows PCR products obtained from 3C library of fixed chromatin and second panel displays the amplified products from randomly ligated gDNA from un-crosslinked chromatin. The intensity of the bands fades out as the distance in kbs increases from locus 2 towards Dhcβ C-2 on chromosome 5 and increases at locus B (approx. 800 kb from HspN) and locus B plus 3alt close to Dhcβ C-2. This was not because of differing primer efficiencies as 2F displays equal primer efficiency with primers for other loci (Fig 4B, panel 3). Thus, ligated 3C library is enriched in interacting locus 2 and locus B plus 3alt hybrid. On the other hand, the un-cross-linked control gDNA shows randomly varying intensities of PCR amplicons obtained with 2F as constant primer and primers from other loci being the variable primer. Our results suggest that dilution of cross-linked chromatin promotes ligation of interacting loci (in the vicinity of HspN and Dhcβ C-2 on chromosome 5). The relative cross-linking frequency was plotted from 3 independent experiments as a function of distance in kb from locus 2 as shown in S13 Fig.

We then performed 3C to investigate physical interaction between HspN on chromosome 5 and Dhcɣ C-1 on chromosome 3. We chose different EcoRI loci on chromosome 3, to investigate physical proximity between HspN on chromosome 5 and Dhcɣ C-1 on chromosome 3 (S14 Fig). Primers were designed flanking these EcoRI recognition sites as shown in S15 Fig. The distance of the different recognition sites from the start codon of Dhcɣ C-1 has also been elaborated in S14 Fig.

Semiquantitative PCR of the 3C library generated from formaldehyde cross-linked chromatin showed enrichment of hybrid formed between locus 2 (close to HspN) and locus G-internal, within Dhcɣ C-1 confirming their physical interaction *in vivo* (Fig 4E and 4F). Thus, our 3C results provide biochemical evidence for co-localization of HspN with HspC (Fig 2), Dhcβ C-2 and Dhcɣ C-1 ORFs (Fig 4) in *Giardia* nucleus.

Despite being located distantly on the same chromosome; HspN, HspC, Dhcβ C-2, Dhcβ C-3 (GL50803_8172) (Fig 2B, top panel also shows enrichment of hybrids formed between locus 2 and locus 6 to 10, Dhcβ C-3 is at a distance of 16 kb from HspC); or on different chromosome like Dhcɣ C-1, their spatial proximity within the nucleus may facilitate accurate stitching of the corresponding pre-mRNAs.

We confirmed our 3C data through DNA FISH using Cy3 labelled probe for HspN and FITC labelled probes for Dhcβ C-2, Dhcβ C-3 and Dhcɣ C-1 genes. Our results show that HspN (chromosome 5) co-localizes with Dhcβ C-2 (chromosome 5) with a co-localization frequency of ~ 0.2, Dhcβ C-3 (chromosome 5) with a co-localization frequency of ~ 0.55 and Dhcɣ C-1 (chromosome 3) with a co-localization frequency of ~ 0.43 (Fig 5). We designed FITC labelled probe for CWP1 gene on chromosome 4 to serve as a negative control. HspN does not co-localize with CWP1 gene as shown in Fig 5A and 5B.

Further, we identified novel S/MAR (Scaffold/matrix associated regions) like DNA elements between the split genes on chromosome 5 of *Giardia lamblia* (S23 Fig). S/MARs are known to play a role in gene regulation and dynamism in chromatin architecture in yeast and higher eukaryotes [19]. Recently, S/MAR like DNA elements were reported to be present in *Giardia* [19]. We have bioinformatically predicted and then experimentally validated the presence of novel S/MARs in the neighbourhood of split genes on chromosome 5 reinforcing the

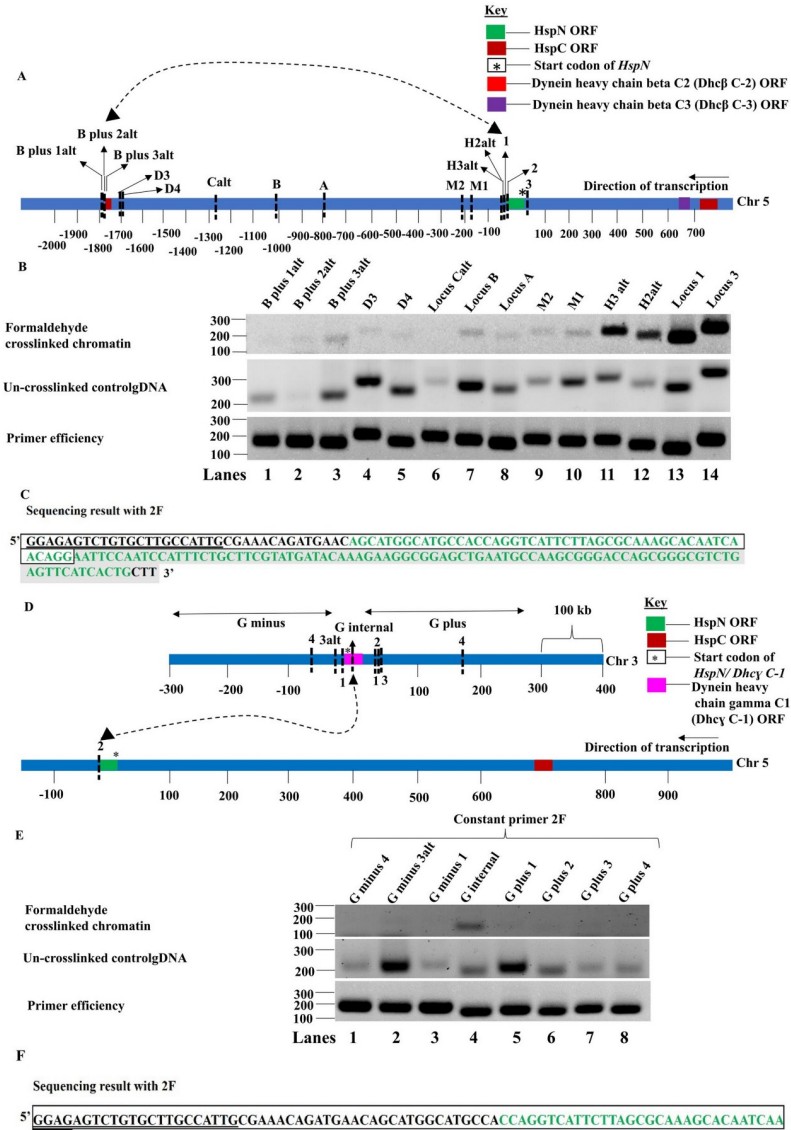

**Fig 4. HspN physically interacts with Dhcβ C-2 and Dhcγ C-1 in the nucleus.** A) Schematic representation of loci chosen to investigate the interaction between HspN and Dhc β C-2 ORFs. B) First panel shows the PCR products obtained with constant primer 2F with other forward primers across the different loci in vicinity and in between HspN and Dhcβ C-2 ORFs. Loci 1 and 3 which are in proximity to locus 2 show intense bands of corresponding sizes. As the distance from locus 2 towards the Dhcβ C-2 increases, the PCR amplicons fade out and the PCR signal then picks up at locus B and then at locus B plus 3alt confirming physical proximity and interaction between loci 2 and B plus 3alt which are proximal to HspN and Dhcβ C-2 ORFs respectively. Second Panel shows randomly obtained hybrids with control gDNA digested and ligated without dilution. Third panel shows PCR amplicons of equal intensities with 2F constant primer and other forward primers indicating equal primer efficiencies. C) The enriched hybrid formed from locus 2 and locus B plus 3alt (Fig 4B, panel 1-lane 3) was excised and subjected to sanger's DNA sequencing with 2F primer. Sequencing result with 2F as the primer could detect the presence of sequences from locus 2 and locus B plus 3alt which confirms the presence of hybrid. E) First panel shows the PCR products obtained with constant primer 2F with other forward primers across the different loci in vicinity and within Dhcγ C-1 ORF on chromosome 3 chosen for this study. Loci within Dhcγ C-1 forms enriched hybrid 3C product with locus 2F in the 3C library preparation. Other loci on either side of Dhcγ C-1 show very faint or no amplification with 2F confirming interaction of HspN with Dhcγ C-1 on chromosome 3. Un-crosslinked control gDNA displays randomly varying intensity of hybrid products indicating that the control template library has randomly ligated products. Third panel shows equal intensities of amplicons indicating equal primer efficiencies. F) The enriched hybrid formed from locus 2 and locus G internal (Fig 4E, panel 1-lane 4) was excised and subjected to sanger's DNA sequencing with 2F primer. Sequencing result with 2F as the primer could detect the presence of sequences from locus 2 and locus G internal which confirms the presence of hybrid.

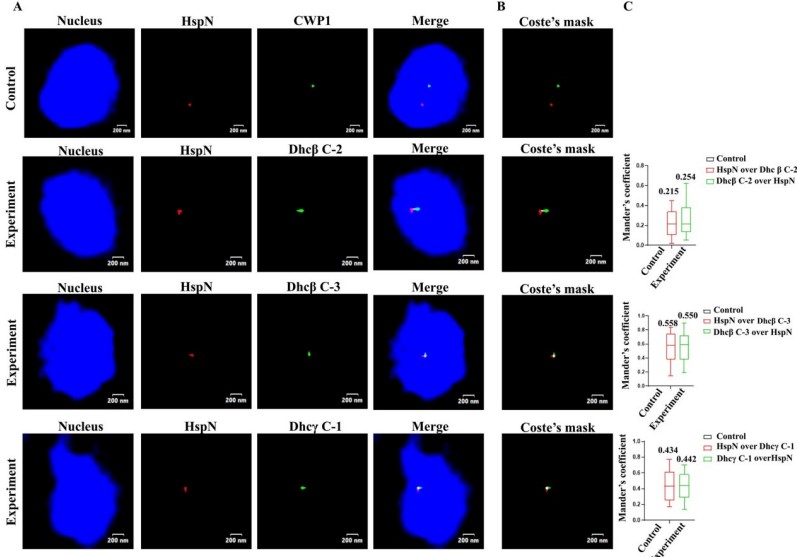

**Fig 5. Fluorescence *in situ* hybridization shows co-localization of HspN and Dynein ORFs within the nucleus.** A) Representative two-coloured FISH images showing localization of HspN on chromosome 5 and Dhcβ C-2 (chromosome 5), Dhcβ C-3 (chromosome 5) and Dhcɣ C-1 (chromosome 3). Top panel shows FISH using probes against HspN (red) and CWP1 (green) which serves as a negative control for co-localization. Merged images represent co-localization of HspN (red) and dynein genes (green). Multiple "Z stacks" were captured for every image in control as well as experiments and the resulting images were stacked to give the "merged" image. Genomic DNA was counterstained with DAPI (blue). B) Depicts the Coste's mask (merge of green and red foci) for representative FISH images. Merged foci (white dot) as shown in experiment depicts true co-localization of the HspN and dynein genes. C) Quantitation of co-localization signal in *Giardia* nuclei. Box-and-whisker plots depicting Mander's co-localization coefficient (range 0–1, where 0 signifies no co-localization and 1 signifies100% co-localization) as evaluated by JACoP plug-in of ImageJ software. A minimum of 30 cells were analysed for co-localization of red and green signals in each experiment and plotted as the fraction of red overlapping green (red) and green overlapping red (green). Control refers to co-localization analysis performed with HspN and CWP1 probe.

role of chromatin topology in facilitating molecular stitching of split genes in *Giardia* nucleus (S23–S33 Figs). Detailed analysis of regulatory aspects of S/MARs in chromatin dynamics is out of the scope of the current manuscript.

Our results suggest a model wherein chromatin looping may facilitate long range contact of HspC, Dhcβ C-2 and Dhcβ C-3 with HspN on chromosome 5. Dhcɣ C-1 on chromosome 3 also co-localizes with HspN on chromosome 5 indicating the possible presence of a specialized molecular hub in *Giardia* nuclei to bring about *trans*-splicing based expression of corresponding split genes (Fig 6).

## Discussion

In our previous study, we have shown that Hsp90 gene of *Giardia* is fragmented into two ORFs (HspN and HspC) separated by 777 kb of linear genetic distance on chromosome 5. Individual pre-mRNAs generated from these split genes undergo post-transcriptional stitching to generate full length Hsp90 RNA by *trans*-splicing reaction [6]. Subsequently, two other essential genes; Dhcβ and Dhcɣ were observed to be expressed by similar *trans*-splicing based mechanism[7–9]. In this study, we have focussed on the mechanism supporting this unique splicing process by examining a) nuclear factor(s) associating with the precursor mRNAs b) nuclear organization of HspN and HspC genes.

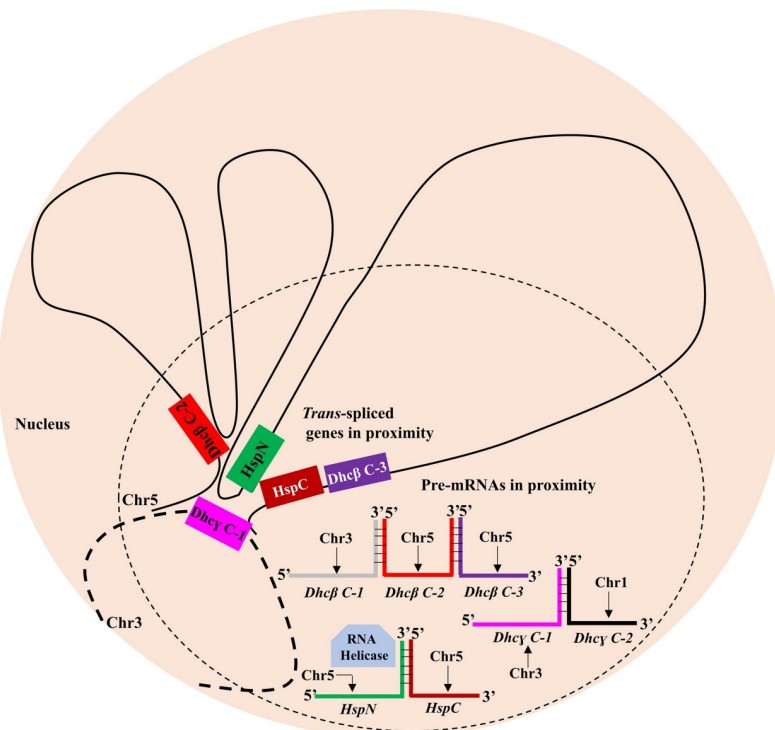

**Fig 6. Model for mechanism of *Giardia* gene *trans*-splicing *in vivo*.** Chromatin looping establishes long range contact between the *trans*-spliced genes in *Giardia lamblia* precisely between locus 2 and loci 6–11 in the vicinity of HspN and HspC ORFs; between locus 2 and locus B plus 3alt in the vicinity of HspN and Dhcβ C-2 and locus 2 and locus G internal close to HspN and within Dhcγ C-1. The pre-mRNAs independently transcribed from the respective ORFs in proximity, thus would be in proximity to facilitate accurate *trans*-splicing of split genes aided by nuclear factors like RNA helicase.

Given that canonical sequence elements such as i) 5' splice site-GU, ii) 3' splice site- AG, iii) branch point adenine and iv) polypyrimidine tract present in cis-spliced RNA were found in split Hsp90, we explored the possibility that there would be mechanistic similarities in their splicing involving spliceosomal components and accessory factors. *Giardia* has minimal spliceosomal machinery as well as canonical nucleotide signatures in 6 cis-spliced genes and 3 *trans*-spliced genes identified so far [3–6,8,9]. Previous bioinformatic analysis has revealed the presence of minimal spliceosomal machinery; the conserved snRNAs and major snRNA-associated proteins in *Giardia* [3,26]. Further, Vanessa Gomez *et al.*, showed the *in vivo* expression of core spliceosomal proteins SmB, SmD3, SmD1, SmD2, SmE, SmF in a complex with U1, U2 and U4 snRNAs [13]. The presence of core snRNAs and snRNPs, introns and the spliceosomal proteins or its homologues indicates the likelihood of a functional spliceosome in *Giardia*.

Our RNA-pull down experiments involving HspN pre-mRNA, HspC pre-mRNA and full-length RNA revealed RNA binding proteins selectively binding to HspN and not to HspC or the full length Hsp90 RNA. Through mass spectrometry we identified a putative DEAD box RNA helicase selectively interacting with HspN pre-mRNA along with other RNA binding protein(s) found to be associating with the pre-mRNAs. It appears likely that the RNA helicase may play a role in unwinding the secondary structure in the complementary sequence of *HspN* (S4 Fig) making it available for interaction with the complementary sequence in *HspC*. ATP dependent RNA helicase DHR1 has been reported to be downregulated during the developmental switch of *Giardia* trophozoites to cysts and has possible role in the complex post-

transcriptional regulation during encystation [27]. Our RNA-protein pull down result suggests another possible role of DHR1 ATP dependent RNA helicase, in the post-transcriptional stitching of *HspN* and *HspC* which remains to be investigated. It is very likely that more than one RNA helicases may be involved in the trans-splicing reaction described in this study. We hope to describe involvement of additional helicases and their functional roles in future studies. Our results highlight that molecular stitching of independent Hsp90 pre-mRNAs from split genes is mediated by spliceosomes *in vivo* and RNA helicase may act as an RNA chaperone to further facilitate *trans*-splicing.

Regulation of eukaryotic gene expression is driven by intricate molecular mechanisms dictated by chromatin structure and nuclear organization [28]. Dynamically changing chromatin looping allows non-random, long-range interactions between genes/DNA elements to fine tune their expression [29]. Various studies have substantiated the role of chromatin loops to accomplish long range contacts between genes and their regulatory elements often positioned several thousand kbs apart on the same or different chromosomes [30].

Despite being separated by 777 kb, HspN and HspC genes are likely to be spatially proximal in *Giardia* nucleus to facilitate corresponding mRNAs to find each other for post transcriptional stitching. Indeed, our 3C experiments indicated that upon crosslinking of chromatin from *Giardia* nucleus, loci at a distance of 7.9 kb from *HspN* and loci at a distance of 30 kb from *HspC* exhibit spatial proximity in the nucleus. On the other hand, locus close to HspN fail to interact with unrelated genes (Protein 21.1, MCM2, Enolase, basal body protein in the vicinity of locus 31) on chromosome 5.

Our study implicates the functional relevance of long-range interaction between HspN and HspC employing intricate chromatin looping mechanism. A recent study revealed the importance of chromatin looping along with CTCF in recruitment of alternate exons for alternative splicing [31]. Clearly chromosome looping between HspN and HspC facilitated by chromatin architecture on chromosome 5 played a role in bringing corresponding genes in close juxtaposition. Our experiments enriching SAR and MAR in the neighbourhood of split genes on chromosome 5 further support this view about nuclear organization.

The spatial juxtapositioning of HspN and HspC suggested by Chromosome conformation capture technology was corroborated by direct microscopic visualization by DNA-FISH experiment. DNA FISH demonstrated co-localization of fluorescent signals from HspN and HspC gene loci with an average co-localization frequency of 0.75 among the 30 nuclei analyzed. However, two other genes, namely Enolase and Protein 21.1 located in between HspN and HspC on the same chromosome showed no co-localization. Additionally, HspN probe failed to co-localize with another unrelated gene CWP1 on a different chromosome, further validating that HspN and HspC are in 3D spatial proximity.

Further our 3C and DNA FISH experiments of HspN with other heterologous split genes, Dhc β C-2, Dhc β C-3 on chromosome 5 and Dhcɣ C-1 on chromosome 3 also suggested the possibility that the other *trans*-spliced genes may occupy a common sub-compartment in *Giardia* nucleus which may serve as a hub for various chromatin-RNA transactions. It appears possible that chromatin structure may bring the *trans*-spliced genes in close proximity in a specialized region enriched in factors necessary for *trans*-splicing to take place.

In addition to chromatin looping mechanism, other specific RNA binding proteins may participate to ensure fidelity and accuracy of the *trans*-splicing reaction *in vivo*. It is plausible that a sub-compartment containing all the split genes, accessory proteins and spliceosomal factors may be present as a phase separated entity to ensure accurate repair and regulation of *trans*-splicing-based expression of split genes in *Giardia lamblia*.

Our attempt to understand the *in vivo* mechanism of Hsp90 *trans*-splicing could indeed be useful to further expand our understanding of the molecular mechanisms of this elegant

process. In addition, the results from our study could be leveraged to design better anti-giardial drugs to combat *Giardiasis*, which remains to be a public health concern in developing and under-developed countries.

## Supporting information

**S1 Fig. HspN and HspC wild type partial sequences; and FLHsp90 sequences cloned in pRSET-C vector.** A, B) Underlined residues indicate the nucleotides spliced out after *in vitro* transcription and incubation in *trans*-splicing (ts) buffer. Nucleotides in bold are the complementary sequence elements in both the partial sequences resulting in pre-mRNAs harbouring complementary nucleotides after *in vitro* transcription. The nucleotides highlighted in the sequences represent the critical nucleotides, 5' SS-GT in HspN clone (GU in HspN pre-mRNA); 3'SS-AG, polypyrimidine tract CCTCCCT (CCUCCCU in HspC pre-mRNA) and branch point Adenine in HspC clone, C) Sequence of FLHsp90 with the *trans*-spliced junction is demarcated. FLHsp90 clone after *in vitro* transcription would generate FLRNA of 947 nts which served as positive control for *in vitro trans*-splicing assay.
(TIF)

**S2 Fig. Strategy for preparation of pre-mRNAs (*HspN, HspC, HspNΔ26, HspCΔ27*) without multiple cloning site (MCS) in the vector.** Underlined residues indicate the intronic regions of pre-mRNAs. Nucleotides in bold are the complementary sequence elements. The nucleotides highlighted in the sequences represent the critical nucleotides; splice sites 5'SS-GT and 3'SS-AG. Also, sequences highlighted at the beginning and at the end indicate forward and reverse primers used to generate corresponding pre-mRNAs lacking the MCS. Forward primer in each case begins with T7 promoter sequence (5'-TAATACGACTCACTATAGGG-3') with 3 additional Gs at the 3' end before the target specific sequence. S2A and S2B Fig indicate precisely the Primer design strategy for HspN and HspC pre-mRNAs to be employed for body labelling of the pre-mRNAs with biotin-16-UTP for the *in-vitro* RNA-protein pull down assay. Same primer sets were used to generate *HspNΔ26* and *HspCΔ27* from clones previously cloned lacking the complementary sequences [23].
(TIF)

**S3 Fig. Quality check of the *in vitro* transcribed biotinylated wild type and mutant HspN and HspC pre-mRNAs on 1.5% MOPS-formaldehyde agarose gel.** Linearized wild type and mutant plasmids were subjected to PCR using primer having T7 promoter sequence, the amplified product was gel purified and subjected to *in vitro* transcription and resolved on MOPS-formaldehyde agarose gel to check the integrity of the RNAs. All of the pre-mRNAs, mutant and wild type, were intact with a single compact band running at the corresponding sizes and hence were used for pull-down assay.
(TIF)

**S4 Fig. Centroid model for partial HspN and HspC pre-mRNAs (Sequence shown in S1 Fig).** A) Stem loop enlarged in the inset harbors the 26 nt complementary functional sequence in partial HspN pre-mRNA. B) Stem loop enlarged in the inset harbors the 26 nt complementary sequence, branch point adenine and polypyrimidine tract functional sequences in partial HspC pre- mRNA.
(TIF)

**S5 Fig. Workflow for chromosome conformation capture (3C) and quality check of cross-linked chromatin and un-crosslinked control gDNA.** A) Schematic representation of 3C workflow. The chromatin conformation in the nuclei of $0.5 \times 10^8$ log phase *Giardia*

trophozoites were fixed with 1.5% formaldehyde. Same number of cells ($0.5 \times 10^8$) were not treated with formaldehyde and served as un-crosslinked control. Nuclei were isolated from both cross-linked and un-crosslinked cells. Chromatin was obtained from formaldehyde fixed cells by lysing the nuclear membrane. Crosslinked chromatin was then digestion with EcoR1. Digestion was followed by dilution (4 folds) to promote intramolecular ligation of cross-linked chromatin over intermolecular ligation to determine interacting loci in the fixed chromatin. Ligation was followed by reversal of crosslinks and identification of interacting loci on the chromatin using locus specific primers by semi-quantitative polymerase chain reaction. Genomic DNA was extracted from control un-crosslinked nuclei, digested with EcoR1 and ligated without dilution to promote random ligations. B) Agarose gel profile of intact, digested and ligated gDNA from un-crosslinked control chromatin and formaldehyde cross-linked chromatin. Lanes 1 and 4 show the intact control gDNA and formaldehyde fixed chromatin, lanes 2 and 5 show digested control gDNA and cross-linked chromatin; and lanes 3 and 6 show randomly ligated control gDNA without dilution and formaldehyde cross-linked chromatin post dilution.
(TIF)

**S6 Fig. Principle of chromosome conformation capture (3C).** Black lines in the enlarged digested species represents chromatin. Green rectangle represents HspN ORF, red rectangle represents HspC ORF and black oval represents proteins bound to the chromatin. A) Formaldehyde cross-linked chromatin upon digestion is subjected to several fold dilution. Upon dilution, the digested molecules remain constant however, the volume would increase and therefore the intermolecular distance between the different digested species. When T4 DNA ligase is introduced at a particular dilution when the intermolecular distance increases beyond 200 nm, intramolecular ligation would be preferred over intermolecular ligation. 3C technology relies on detecting the hybrids that are enriched in the 3C library. B) Un-crosslinked extracted *Giardia* genomic DNA was utilized for negative control experiment. When genomic DNA was purified from the proteins, the DNA was not constrained in any structure. However, it was only contiguous stretch of nucleotides without proteins constraining it into any structure. The extracted genomic DNA was digested and ligated using T4 DNA ligase without dilution. This would result in every digested species having equal probability of ligating with any other digested species in close proximity. Therefore, ligation without dilution would promote random ligation between the digested species in a small volume as opposed to intramolecular ligation promoted in the case of the crosslinked chromatin.
(TIF)

**S7 Fig. Rationale for detecting different hybrids in the 3C library.** Schematic representation of primer design to detect different hybrids between loci. The blueish green rectangle represents chromatin in the vicinity of HspN and the yellowish red rectangle represents chromatin in the vicinity of HspC. The vertical dotted lines represent the EcoRI recognition sites close to HspN and HspC. For simplicity, only two EcoRI recognition sites/two EcoRI loci have been illustrated. However, to examine long range interaction between the two ORFs, 13 different loci (including the two recognition sites shown) were studied. Primers were designed flanking the EcoRI recognition sites. F1, R1 and F2, R2 represents flanking primers close to the EcoRI sites in the vicinity of HspN and HspC. Upon digestion, dilution and ligation of the chromatin different segments; 1, 2, 3 and 4 could ligate to each other in different combinations. Segment 1 could ligate to 3 and/or 4. Segment 2 could ligate to 3 and/or 4. To detect chromatin hybrids formed between segment 1 with segment 3 as shown, we used F1 and F2 as the primers and examined its enrichment using semi-quantitative PCR.
(TIF)

**S8 Fig. Schematic representation of different EcoRI recognition sites/EcoRI loci to examine long range interaction between HspN and HspC on chromosome 5, Figure made to scale.** A) The different vertical lines represent the different EcoRI sites chosen for examining physical proximity of HspN and HspC on chromosome 5. The EcoRI sites are designated uniquely e.g., 1, 2, 3, 4, 5, 40, 31 etc., which are indicated above each vertical lines. The key on right top indicates the genes under investigation, the EcoRI loci chosen for study highlighted with the scissor symbol and indicates the start codon of HspN. B) The table displays the distances of each EcoRI loci chosen for the study from the start codon of HspN. The minus (-) symbol denotes that the distance in bp is towards the left of HspN and the plus (+) symbol denotes that the distance in bp is towards the right of HspN.
(TIF)

**S9 Fig. Schematic representation displaying the forward and reverse primers flanking each EcoRI site chosen to examine physical proximity between HspN and HspC on chromosome 5.** The numbers 1,2,3,4,5,40, 31, 6,7,8,9,10,11 represent the different EcoRI loci/EcoRI recognition sites which were investigated. The numbers below the map (-100, 200,300. . .) serves as a ruler and shows the distances of the different EcoRI loci from each other and from the genes under question. The red arrows represent the forward primers, and the black arrows represent the reverse primers.
(TIF)

**S10 Fig. HspN and HspC ORFs physically interact in the nucleus.** A) The three panels show the PCR products obtained with constant primer 2F with other forward primers designed across the different loci in vicinity and in between HspN and HspC ORFs chosen for this study. Loci 1 and 3 which are in proximity to locus 2 show intense bands of corresponding sizes. As the distance from locus 2 increases, the PCR amplicons fade out and the PCR signal then starts picking up from locus 6 gradually; increasing significantly at locus 11 confirming physical proximity and interaction between loci 2 and loci 11 which are proximal to HspN and HspC ORFs respectively. Similar trend of intensities was observed of the different hybrids in the 3C libraries from three different 3 experiments. C) The relative crosslinking frequency of locus 2 proximal to HspN was determined from the three experiments and plotted as a function of distance from the start codon of HspN.
(TIF)

**S11 Fig. Schematic representation of different EcoRI recognition sites/EcoRI loci to examine long range interaction between HspN and Dhcβ C-2 on chromosome 5, Figure made to scale.** A) The different vertical lines represent the different EcoRI sites chosen for examining physical proximity of HspN and Dhcβ C-2 on chromosome 5. The EcoRI sites are designated uniquely e.g., B plus 1alt, B plus 2alt, B plus 3alt etc., which are indicated above each vertical line. The key on right top indicates the genes under investigation, the EcoRI loci chosen for study highlighted with the scissor symbol and the key also indicates the start codon of HspN. The numbers below the map (-2000, -1900, -1800, -1700 and so on) serves as a ruler and shows the distances of the different EcoRI loci from each other and from the genes under question. B) The table displays the distances of each EcoRI loci chosen for the study from the start codon of HspN. The minus (-) symbol denotes that the distance in bp is towards the left of HspN and the plus (+) symbol denotes that the distance in bp is towards the right of HspN.
(TIF)

**S12 Fig. Schematic representation displaying the forward and reverse primers flanking each EcoRI site chosen to examine physical proximity between HspN and Dhcβ C-2 on chromosome 5.** The red arrows represent the forward primer and the black arrows represent the reverse primers. Please refer to S11 Fig to correlate EcoRI recognition sites with their corresponding names.
(TIF)

**S13 Fig. HspN physically interacts with Dhcβ C-2 in the nucleus.** A) The panels show the PCR products obtained with constant primer 2F with other forward primers across the different loci in vicinity and in between HspN and Dhcβ C-2 ORFs. Loci 1 and 3 which are in proximity to locus 2 show intense bands of corresponding sizes. As the distance from locus 2 towards the Dhcβ C-2 increases, the PCR amplicons fade out and the PCR signal then picks up at locus B and then at locus B plus 3alt confirming physical proximity and interaction between loci 2 and B plus 3alt which are proximal to HspN and Dhcβ C-2 ORFs respectively. Similar trend of intensities was observed of the different hybrids in the 3C libraries from three different 3 experiments. C) The relative crosslinking frequency of locus 2 and locus B plus 3alt proximal to HspN and Dhcβ C-2 ORFs was determined from the three experiments and plotted as a function of distance from locus 2.
(TIF)

**S14 Fig. Schematic representation of different EcoRI recognition sites/EcoRI loci to examine long range interaction between HspN and Dhcγ C-1 on chromosome 3, Figure made to scale.** A) The different vertical lines represent the different EcoRI sites chosen for examining physical proximity of HspN on chromosome 5 and Dhcγ C-1 on chromosome 3. The EcoRI sites are designated uniquely e.g., G minus 4, G minus 3alt, G minus 1 etc., which are indicated above each vertical line. The key on right top indicates the genes under investigation, the EcoRI loci chosen for study highlighted with the scissor symbol and the key also indicates the start codon of HspN and Dhcγ C-1. The numbers below the map of chromosome 3 (-300, -200, -100 and so on till 400) serve as ruler to show the relative genomic distances between the EcoRI loci on chromosome 3. B) The table displays the distances of each EcoRI loci chosen for the study from the start codon of Dhcγ C-1. The minus (-) symbol denotes that the distance in bp is towards the left of Dhcγ C-1 gene and the plus (+) symbol denotes that the distance in bp is towards the right of Dhcγ C-1 gene.
(DOCX)

**S15 Fig. Schematic representation displaying the forward and reverse primers flanking each EcoRI site chosen to examine physical proximity between HspN on chromosome 5 and Dhcγ C-1 on chromosome 3.** Schematic representation displaying the forward and reverse primers flanking each EcoRI site chosen to examine physical proximity between HspN on chromosome 5 and Dhcγ C-1 on chromosome 3. The Figure shows the EcoRI recognition sites on chromosome 3. The red arrows represent the forward primers, and the black arrows represent the reverse primers.
(TIF)

**S16 Fig. HspN physically interacts with Dhcγ C-1 in the nucleus.** A) The panels show the PCR products obtained with constant primer 2F with other forward primers across the different loci in vicinity and within Dhcγ C-1 ORF on chromosome 3 chosen for this study. Loci within Dhcγ C-1 forms enriched hybrid 3C product with locus 2F in the 3C library preparation. Other loci on either side of Dhcγ C-1 show very faint or no amplification with 2F confirming interaction of HspN with Dhcγ C-1 on chromosome 3. Similar trend of intensities was observed of the different hybrids in the 3C libraries from three different 3 experiments. B) The

relative crosslinking frequency of locus 2 and locus internal to Dhcγ C-1 (G internal) along with the loci chosen upstream and downstream of Dhcγ C-1 was determined from the three experiments and plotted as a function of distance from the start codon of Dhcγ C-1.
(TIF)

**S17 Fig. Sequencing result after 3C PCR using 2F primer confirms the presence of enriched hybrid between HspN and HspC ORFs confirming their interaction in the nucleus.** The enriched hybrid formed from locus 2 and locus 11 (Fig 2B, panel 1-lane 12) was excised and subjected to Sanger's DNA sequencing with 2F primer. Sequencing result with 2F as the primer could detect the presence of sequences from locus 2 and locus 11 which confirms the presence of this hybrid in the enriched band.
(TIF)

**S18 Fig. Sequencing result after 3C PCR using 11F primer confirms the presence of enriched hybrid between HspN and HspC ORFs confirming their interaction in the nucleus.** The enriched hybrid formed from locus 2 and locus 11 (Fig 2B, panel 1-lane 12) was excised and subjected to Sanger's DNA sequencing with 11F primer. Sequencing result with 11F as the primer could detect the presence of sequences from locus 11 and locus 2 which confirms the presence of the hybrid in the enriched band.
(TIF)

**S19 Fig. Sequencing result after 3C PCR using 2F primer confirms the presence of enriched hybrid between HspN and Dhcβ C-2 ORFs confirming their interaction in the nucleus.** The enriched hybrid formed from locus 2 and locus B plus 3alt (Fig 4B, panel 1-lane 3) was excised and subjected to sanger's DNA sequencing with 2F primer. Sequencing result with 2F as the primer could detect the presence of sequences from locus 2 and locus B plus 3alt which confirms the presence of hybrid.
(TIF)

**S20 Fig. Sequencing result after 3C PCR using B plus 3alt-F primer confirms the presence of enriched hybrid between HspN and Dhcβ C-2 ORFs confirming their interaction in the nucleus.** The enriched hybrid formed from locus 2 and locus B plus 3alt (Fig 4B, panel 1-lane 3) was excised and subjected to sanger's DNA sequencing with B plus 3alt-F primer. Sequencing result with B plus 3alt-F as the primer could detect the presence of sequences from locus B plus 3alt and locus 2 which confirms the presence of hybrid.
(TIF)

**S21 Fig. Sequencing result after 3C PCR using 2F primer confirms the presence of enriched hybrids between HspN and Dhcγ C-1 ORFs confirming their interaction in the nucleus.** The enriched hybrid formed from locus 2 and locus G internal (Fig 4E, panel 1 lane 4) was excised and subjected to sanger's DNA sequencing with 2F primer. Sequencing result with 2F as the primer could detect the presence of sequences from locus 2 and locus G internal which confirms the presence of hybrid.
(TIF)

**S22 Fig. Sequencing result after 3C PCR using G internal-F primer confirms the presence of enriched hybrids between HspN and Dhcγ C-1 ORFs confirming their interaction in the nucleus.** The enriched hybrid formed from locus 2 and locus G internal (Fig 4E, panel 1 lane 4) was excised and subjected to sanger's DNA sequencing with G internal-F primer. Sequencing result with G internal-F as the primer could detect the presence of sequences from locus G internal and locus 2 which confirms the presence of hybrid.
(TIF)

**S23 Fig. Marscan predicts the presence of S/MAR like elements between *trans*-spliced genes on chromosome 5.** A) S/MAR like DNA sequence elements predicted by marscan across all the chromosomes of *Giardia lamblia* assemblage A WB C6. *G. lamblia* genomic DNA was investigated for the presence of S/MAR rules reported by Singh *et al*., 2000. Marscan online tool as well as in-house script predicted 20 S/MAR like DNA elements across all the 5 chromosomes of *Giardia*. B) Marscan predicts the presence of potential S/MAR like element in between HspN and HspC ORFs. Simplified schematic displays the presence of all the S/MAR rules i) bipartite motif, ii) general AT richness, iii) ORI site, iv) topoisomerase binding site, v) kinked DNA followed by a 204 bp and 725 bp region between HspN and HspC ORFs on chromosome 5 [32]. C) Schematic representation of the distribution of S/MARs predicted bioinformatically on Chromosome 5 in between the *trans*-spliced genes.
(TIF)

**S24 Fig. Analysis of enriched MARs confirms the presence of *in silico* predicted S/MAR like elements between *trans*-spliced genes on chromosome 5.** A) Schematic representation of the workflow for enrichment of S/MAR like DNA elements from *G. lamblia*. B) Size distribution and identity of enriched S/MAR DNA on 1.2% agarose gel. Lane 1 shows intact genomic DNA from *Giardia lamblia*; lane 3 shows enriched S/MAR DNA ranging in size upto approximately 200 bps. Lane 5 shows enriched S/MAR DNA elements post DNAse treatment; lane 7 shows S/MAR DNA element post RNase treatment. C) PCR validates the presence of bioinformatically predicted S/MAR elements between HspN and HspC ORFs. Top panel shows PCR products obtained with the genomic DNA as the template. Lower panel displays PCR validation of predicted S/MARs. Lane 1 shows amplicon of 106 bps with internal primers for the predicted 204 bp S/MAR element, lane 2 shows amplicon of 204 bps with primers designed at the ends of the predicted S/MAR sequence, lane 3 shows the PCR product of another predicted S/MAR of 108 bps, Lane 4 shows presence of previously reported S/MAR by Padmaja *et al* [19]; thus serves as the positive control. Lane 5 serves as the negative control. D) PCR validates the presence of bioinformatically predicted S/MAR elements between HspN and Dhcβ-C2. Top panel shows PCR products obtained with the genomic DNA as the template. Lower panel displays PCR validation of predicted S/MARs. Lane 1 shows amplicon of 163 bps with primers for the predicted S/MAR element between HspN and Dhcβ C-2, lane 2 shows amplicon of 202 bps with primers designed for another predicted S/MAR sequence, lane 3 shows the PCR product obtained with internal primers for S/MAR Dyn 4 between HspN and Dhcβ C-2, Lane 4 shows presence of previously reported S/MAR, thus serves as the positive control. Lane 5 serves as the negative control.
(TIF)

**S25 Fig. Sequencing result confirms the presence of S/MAR element(s) between HspN and HspC on chromosome 5.** Figure shows Sanger's DNA sequencing results with S/MAR 90 1-F primer of band excised from lane 2 (Bottom panel) in S24C Fig.
(TIF)

**S26 Fig. Sequencing result confirms the presence of S/MAR element(s) between HspN and HspC on chromosome 5.** Figure shows Sanger's DNA sequencing results with S/MAR 90 1-R primer of band excised from lane 2 (Bottom panel) in S24C Fig.
(TIF)

**S27 Fig. Sequencing result confirms the presence of S/MAR element(s) between HspN & Dhcβ C-2 on chromosome 5.** Figure shows sequencing results with S/MAR Dyn3-F primer of band excised from lane 1 (Bottom panel) in S24D Fig.
(TIF)

**S28 Fig. Sequencing result confirms the presence of S/MAR element(s) between HspN & Dhcβ C-2 on chromosome 5.** Figure shows sequencing results with S/MAR Dyn3-R primer of band excised from lane 1 (Bottom panel) in S24D Fig.
(TIF)

**S29 Fig. Sequencing result confirms the presence of S/MAR element(s) between HspN & Dhcβ C-2 on chromosome 5.** Figure shows sequencing result with S/MAR Dyn4-F primer of bands excised from lane 2 (Bottom panel) in S24D Fig.
(TIF)

**S30 Fig. Sequencing result confirms the presence of S/MAR element(s) between HspN & Dhcβ C-2 on chromosome 5.** Figure shows sequencing result with S/MAR Dyn4-R primer of bands excised from lane 2 (Bottom panel) in S24D Fig.
(TIF)

**S31 Fig. Sequencing result confirms the presence of S/MAR element(s) between HspN & Dhcβ C-2 on chromosome 5.** Figure shows sequencing result with S/MAR Dyn4 internal-F primer of band excised from lane 3 in S24D Fig.
(TIF)

**S32 Fig. Sequencing result for the positive control band excised from lane 4 (Bottom panel) of S24C Fig.** Figure shows DNA sequencing result of the positive control with S/MAR positive control-F primer.
(TIF)

**S33 Fig. Sequencing result for the positive control band excised from lane 4 (Bottom panel) of S24C Fig.** Figure shows DNA sequencing result of the positive control with S/MAR positive control-R primer.
(TIF)

**S1 Data. List of primer sequences used in the study.**
(XLSX)

## Acknowledgments

Authors thank Prof. Didier Picard and Prof. Katharina Strub from University of Geneva for their inputs and suggestions during initial stages of this study.

## Author Contributions

**Conceptualization:** Vinithra Iyer, Rakesh Mishra, Utpal Tatu.

**Data curation:** Vinithra Iyer, Rakesh Mishra.

**Formal analysis:** Vinithra Iyer, Sheetal Tushir, Shreekant Verma, Srimonta Gayen, Rakesh Mishra, Utpal Tatu.

**Funding acquisition:** Utpal Tatu.

**Investigation:** Utpal Tatu.

**Methodology:** Vinithra Iyer, Sheetal Tushir, Shreekant Verma, Sudeshna Majumdar, Srimonta Gayen, Utpal Tatu.

**Resources:** Srimonta Gayen.

**Supervision:** Utpal Tatu.

**Validation:** Vinithra Iyer, Sheetal Tushir, Rakesh Mishra, Utpal Tatu.

**Visualization:** Sheetal Tushir, Sudeshna Majumdar.

**Writing – original draft:** Vinithra Iyer.

**Writing – review & editing:** Vinithra Iyer, Sheetal Tushir, Utpal Tatu.

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
