## [Decision Letter · Decision Letter 0]

23 May 2021

Dear Prof. Tatu,

Thank you very much for submitting your manuscript "The role of nuclear organization in trans-splicing based expression of Heat shock protein 90 in Giardia lamblia" for consideration at PLOS Neglected Tropical Diseases. As with all papers reviewed by the journal, your manuscript was reviewed by members of the editorial board and by several independent reviewers. In light of the reviews (below this email), we would like to invite the resubmission of a significantly-revised version that takes into account the reviewers' comments. 

We cannot make any decision about publication until we have seen the revised manuscript and your response to the reviewers' comments. Your revised manuscript is also likely to be sent to reviewers for further evaluation.

Sincerely,

Renata Rosito Tonelli, PhD

Associate Editor

Anthony Papenfuss

Deputy Editor

Reviewer's Responses to Questions

**Key Review Criteria Required for Acceptance?**

**Methods**

-Are the objectives of the study clearly articulated with a clear testable hypothesis stated?

-Is the study design appropriate to address the stated objectives?

-Is the population clearly described and appropriate for the hypothesis being tested?

-Is the sample size sufficient to ensure adequate power to address the hypothesis being tested?

-Were correct statistical analysis used to support conclusions?

-Are there concerns about ethical or regulatory requirements being met?

Reviewer #1: Statistical presentation incorrect - should show individual results when only 3 samples used. Most definitely not the SEM.

Reviewer #2: The objectives and hypothesis are clear and the method do address the hypothesis.

However the study design still misses some silencing/knockdown experiments of the DEAD/DEXH-box RNA helicase to ensure its participation in the process.

**Results**

-Does the analysis presented match the analysis plan?

-Are the results clearly and completely presented?

-Are the figures (Tables, Images) of sufficient quality for clarity?

Reviewer #1: Figures 2 and 4 are incomprehensible.

Reviewer #2: The analysis presented matches well the planned assays.

The results are clear, and some of them even elegant in design. However, since the DEAD/DEXH-box RNA helicase was isolated from a gel and the authors did not state that the corresponding area of the gels in the deleted construct were devoid of such protein, additional support for the involvement of this RNA helicase is needed. 

The graphical quality of the images must be improved since they appear to be prepared in a rushed manner. Image copy-pasting from the RNA folding programs is not sufficient for clarity. RNA hybrids in the splicing model should be fully written out.

**Conclusions**

-Are the conclusions supported by the data presented?

-Are the limitations of analysis clearly described?

-Do the authors discuss how these data can be helpful to advance our understanding of the topic under study?

-Is public health relevance addressed?

Reviewer #1: Are the conclusions supported by the data presented?

Partly no, and partly I don't know because the Figures are so unclear.

First, the authors do a pull-down using RNA probes, and extracts. They identify an RNA helicase, among other proteins. They claim the interaction is specific but provide no evidence that this is the case. Where are the quantitation, negative controls? They do not provide evidence for binding of the helicase in viivo. There is also no evidence that the helicase plays any role in splicing, or even that it is in the nucleus. RNA Helicases are not generally sequence-specific and are often quite abundant. This is a very weak part of the paper and without additional functional evidence it (and all relevant Figures and tables) should be removed. As far a I can see all the authors have done is to identify a protein from a random part of a polyacrylamide gel.

Figure 2: I think I vaguely understand what the authors are showing here but the Figure is really confusing and so is the Legend. What is an “EcoRI locus”? What are the numbers above and below the map? What are the products obtained with “randomly ligated” DNA? What are the “other forward primers”? What are “different loci chosen for this study”? A much clearer diagram is required showing locations and directionsof all primers used. Also, to verify the results the PCR products must be sequenced. In panel C, with only 3 measurements you cannot legitimately use the standard error of the mean or even the standard deviation. Please show the individual measurements instead. 

Figure 3: This shows that probes that hybridise to HspN and HspC ORFs colocalise. That is interesting, but I am not sure about the negative control gene. Is this gene on the same chromosome? If not, it is not valid as a control. A better control would be to try two different genes that are located in between the HspN and HspC ORFs on the same chromosome. Another control that is essential is to do the same experiment after inhibiting transcription, to see whether the mutual association depends on RNA.

Figure 4 is truly incomprehensible. I gave up trying to understand it. If the conclusion is that the two separate trans spliced loci are all together in one place, this actually seems strange because I would not expect it. (This is not the equivalent of spliced leader trans splicing, and most trypanosomes genes are also not located near the spliced leader array.) In any case, the results need to be be confirmed by sequencing and by in situ hybridisation, again with additional appropriate controls. 

Any discussion is premature because this needs more experimental work in order to support the conclusions.

Reviewer #2: For the most part, conclusions are supported, but additional support for the DEAD/DEXH-box RNA helicase is needed. 

As stated before, the limitations are not clearly described on this issue alone.

The authors do discuss the importance and relevance of their study, including public health issues.

**Editorial and Data Presentation Modifications?**

Reviewer #1: The English needs correction, articles (“The” and ‘a”) are pervasively missing.

Lines 72-74 - why do some species have capital letters and others not? None should, and trans splicing doesn’t need a capital letter either. Trans should be italicised, though.

Line 323 - “dispersed” is the wrong word. it should be “separated”.

Accession numbers for sequences not generated in the paper can be mentioned in the text but so not need to be provided in a separate section.

Reviewer #2: In the introduction, reference 12 is not the best choice for the statement: Sequence analysis of HspN and HspC pre-mRNA substrates showed the presnece of cis-sequence elements; GU-AG intron-exon boundry elements, polypyrimidine tract as well as branch point adenine in Hsp90 pre-mRNAs.

**Summary and General Comments**

Reviewer #1: If the colocalization is true it would be interesting, but more experimental data are needed .

Reviewer #2: This study is highly relevant to the field of co-transcriptional splicing in Giardia. The authors cleverly used 3C assays to understand in vivo transcription and trans-splicing of the Hsp90 separate transcripts with the aid of an RNA helicase. This work provides the first glimpse to nuclear events in Giardia with therapeutic potential.

PLOS authors have the option to publish the peer review history of their article (what does this mean?). If published, this will include your full peer review and any attached files.

Reviewer #1: No

Reviewer #2: Yes: Jesús Valdés
---

## [Decision Letter · Decision Letter 1]

13 Sep 2021

Dear Prof. Tatu,

We are pleased to inform you that your manuscript 'The role of nuclear organization in trans-splicing based expression of Heat shock protein 90 in Giardia lamblia' has been provisionally accepted for publication in PLOS Neglected Tropical Diseases.

Best regards,

Renata Rosito Tonelli, PhD

Associate Editor

Anthony Papenfuss

Deputy Editor

Reviewer's Responses to Questions

**Key Review Criteria Required for Acceptance?**

**Methods**

-Are the objectives of the study clearly articulated with a clear testable hypothesis stated?

-Is the study design appropriate to address the stated objectives?

-Is the population clearly described and appropriate for the hypothesis being tested?

-Is the sample size sufficient to ensure adequate power to address the hypothesis being tested?

-Were correct statistical analysis used to support conclusions?

-Are there concerns about ethical or regulatory requirements being met?

Reviewer #2: The problem and objectives were clearly stated. The methods used are appropriate to test the hypothesis and reach their conclusions.

**Results**

-Does the analysis presented match the analysis plan?

-Are the results clearly and completely presented?

-Are the figures (Tables, Images) of sufficient quality for clarity?

Reviewer #2: The authors have corrected all the issues raised in their first manuscript and the results now appear clean and concise showing the supporting findings. The overall structure of the manuscript has improved.

**Conclusions**

-Are the conclusions supported by the data presented?

-Are the limitations of analysis clearly described?

-Do the authors discuss how these data can be helpful to advance our understanding of the topic under study?

-Is public health relevance addressed?

Reviewer #2: Conclusions are supported by the data. The authors fully describe their contribution in the field and in public health.

**Editorial and Data Presentation Modifications?**

Reviewer #2: No modifications are needed. The authors carefully addressed all the concerns previously raised.

**Summary and General Comments**

Reviewer #2: This work is a novel contribution in the involvement of HSp in trans-splicing mechanism of G. lamblia.

PLOS authors have the option to publish the peer review history of their article (what does this mean?). If published, this will include your full peer review and any attached files.

Reviewer #2: **Yes: **Jesús Valdés Flores

---

## [Editor Report · Acceptance letter]

18 Sep 2021

Dear Prof. Tatu,

We are delighted to inform you that your manuscript, "The role of nuclear organization in trans-splicing based expression of Heat shock protein 90 in Giardia lamblia," has been formally accepted for publication in PLOS Neglected Tropical Diseases.

Best regards,

Shaden Kamhawi

co-Editor-in-Chief

Paul Brindley

co-Editor-in-Chief
